# Efficacy of Classic Ear Molding for Neonatal Ear Deformity: Case Series and Literature Review

**DOI:** 10.3390/jcm11195751

**Published:** 2022-09-28

**Authors:** Jeonghoon Kim, Taehee Jo, Jaehoon Choi, Junhyung Kim, Woonhyeok Jeong

**Affiliations:** Department of Plastic and Reconstructive Surgery, Dongsan Medical Center, College of Medicine, Keimyung University, Daegu 1035, Korea

**Keywords:** ear molding, deformational ear anomaly, auricular anomaly, congenital ear anomaly, nonsurgical correction, newborn

## Abstract

Background: We analyzed an original case series of the classic ear-molding method and evaluated the efficacy and complication rate of the method compared to commercial ear-molding products by meta-analysis to draw conclusions on the efficacy of the classic method. Methods: From January 2019 to March 2022, we selected patients who underwent classic ear molding for newborn ear deformities at our institution and reviewed the patient age, treatment time, efficiency and complications. Additionally, the PubMed, EMBASE, and Scopus databases were searched, and meta-analysis (following the PRISMA guidelines) was performed. Results: In the case study, the success rate (excellent and good outcomes) of the classic ear-molding method was 92.6%. The mean age at application and mean duration of application were 5.81 ± 6.09 days and 32.13 ± 7.90 days, respectively. In the systematic review, the classic method group showed a statistically smaller success rate (proportion of 0.79) and statistically smaller complication rate (proportion of 0.05) than the commercial product group (proportion of 0.83). Conclusions: Compared with commercial products, classic ear molding has remarkable and comparable therapeutic effects on neonatal auricular deformities. Additionally, the classic ear-molding method is more suitable for infants with auricular deformities from socioeconomically vulnerable areas. Thus, the classic ear-molding method could be a better option for congenital ear anomalies than commercial ear-molding products.

## 1. Introduction

Neonatal ear anomalies are classified into malformations or deformations [1,2]. Ear malformations involve a partial loss of the skin or cartilage with underdevelopment of any part of the pinna. Ear deformations involve a fully developed but misshaped pinna with no deficiency of skin and cartilage [3]. Deformational anomalies of the ear are categorized into prominent, lidding/lop, conchal crus, Stahl’s, and helical rim deformities [4,5]. Fortunately, these deformational anomalies can be corrected with an ear-molding technique until the early neonatal period. Circulating levels of maternal estrogen and tissue hyaluronic acid, which favor newborn ear malleability, peak during the first 72 h after birth and decline to baseline by 6 weeks of age, when the auricle becomes more elastic and firm [6]. Thus, during this window, when the ear cartilage is malleable, ear anomalies can be corrected by ear molding. Ear molding in infants may eliminate the need for future surgical correction, and many authors have reported the effectiveness of this intervention [1]. The current literature on ear molding demonstrates the efficacy or timing of neonatal ear molding in the correction of deformational ear anomalies. However, no studies have focused on evaluating the efficacy of the classic ear-molding method. In this study, we evaluated the efficacy and complication rate of the classic ear-molding method compared to commercial ear-molding products. For evaluation, we describe an original case series from our institution. We also conducted a systematic literature review to draw conclusions on the efficacy of the classic ear-molding method.

## 2. Materials and Methods

### 2.1. Case Series

#### 2.1.1. Patient Selection

From January 2019 to March 2022, we selected patients who underwent the classic ear-molding method for newborn ear deformities at our institution and analyzed the patient age at treatment, treatment time, efficiency and complications. Pretreatment and post-treatment photographs were assessed by three independent plastic surgeons, who rated the outcome as poor, moderate, good, and excellent. An agreement test was applied. Definitions of grading are explained in Table 1.

#### 2.1.2. Treatment and Evaluation

The classic ear-molding materials are custom-made splints of metal wire, surgical tape, foam, and silicone tape (Figure 1).

Step 1. The hair around the ear was shaved before ear molding (to avoid damage to the skin), and then the skin oil was wiped off with isopropyl alcohol so the cradle could adhere to the skin.

Step 2. The splint was selected and curved according to the desired shape of the helix and was then placed on the anterolateral surface of the auricle.

Step 3. Fixation on the anterolateral surface of the auricle in the groove between the helix and antihelix was achieved with surgical tape. Surgical tape was also used to position the auricle closer to the scalp.

Step 4. The ear was covered with foam to protect the molding materials.

For the first week of treatment, taping was used to counteract the significant resistive forces to expansion resulting from tissue deficiency. After that, the splint was curved according to the desired shape of the helix and then fixed on the anterolateral surface of the auricle. The infants wore the splint for 24 h per day.

When skin complications occurred, we continued the treatment because foam-covered molding materials absorb exudate and improve skin lesions. The splinting stopped only when pressure ulcers occurred.

When we first started treatment, patients visited the clinic to check for complications at three days after ear-molding. After we confirmed that five patients did not have any complications at three days after treatment, infants were scheduled for weekly follow-up routinely to monitor complications and auricle changes and to modulate the devices to obtain better correcting conditions. Splinting was performed continuously until 1 week after normalization of the auricular anatomy was achieved, and it was stopped after a maximum of 6 weeks of treatment even when no correction occurred.

#### 2.1.3. Statistical Analysis

All analyses were performed using SPSS 21.0 software. Variables were analyzed using frequency analysis, descriptive analysis, one-way ANOVA, the Kruskal-Wallis test, and Fisher’s exact test where appropriate. A *p* value of <0.05 was used as the cutoff point for statistical significance.

### 2.2. Systematic Literature Review

#### 2.2.1. Search Protocol

The PubMed, EMBASE, and Scopus databases were systematically searched. The following keywords were used to search the literature: ‘Neonatal ear molding’ OR ‘Ear molding’ OR ‘Earwell’ Or ‘InfantEar’. The preferred reporting items for systematic reviews and meta-analysis (PRISMA) guidelines were followed to perform this systematic review and meta-analysis. An overview of the systematic review process is presented in Figure 2. Initially, we screened all publications by titles and/or abstracts. Next, we assessed eligibility based on full-text articles. Finally, for the included publications, we collected only cases with clear and relevant data, as described above. Consequently, a total of 13 studies were included in this systematic review (Table 2).

#### 2.2.2. Inclusion and Exclusion Criteria

All studies meeting the following criteria were included: (1) English language and (2) full text available. Studies fulfilling the following criteria were excluded: (1) duplicate publications, (2) studies that did not include neonatal ear molding, (3) clinical trials or review articles, and (4) studies reporting unclear data or withdrawn studies.

#### 2.2.3. Statistical Analysis

Data were retrieved by a single author, and the following information was retrieved from the included studies. Thirteen articles were included in the heterogeneity analysis. The successful correction rate and complication rate along with the 95% confidence interval (CI) were estimated. The Q statistic for heterogeneity and the I^2^ index were calculated. Values less than 50% indicated low heterogeneity; other values indicated high heterogeneity. If I^2^ < 50%, we used a fixed-effects model; if I^2^ > 50%, we employed a random-effects model in our meta-analysis.

## 3. Results

### 3.1. Case Series

We recruited 54 patients with a total of 94 ears. Twenty-five patients (41 ears) were male, and twenty-nine (53 ears) were female. The most common deformity was a lop ear (69.1 percent), followed by a constricted ear (12.8 percent). Forty-eight ears (51.1 percent) were graded excellent, thirty-nine (41.5 percent) were graded good, seven (14 percent) were graded moderate, and none were graded poor. The mean age at application was 5.81 ± 6.09 days. The mean duration of application for all patients was 32.13 ± 7.90 days (Table 3). Five of 94 ears (5.3 percent) experienced complications during the treatment period, which included dermatitis and skin excoriations. These skin complications healed spontaneously. Examples of successful correction are illustrated in Figure 3.

#### 3.1.1. Types of Anomalies and Outcomes

The posttreatment outcomes for the five types of ear deformities are shown in Table 4. The success rates (outcome graded as excellent and good) of the classic ear-molding method were 94%, 83%, 86%, 100%, and 100% for lop ear, constricted ear, helical rim abnormality, Stahl’s ear and prominent ear, respectively. The difference in outcome between the types was not statistically significant (*p* = 0.210).

#### 3.1.2. Patient Age and Duration of Application

The mean age at application was 5.81 ± 6.09 days. Within each outcome group, the average age at initiation was 4.58 days in the excellent group, 7.72 days in the good group, and 3.71 days in the moderate group. This was statistically significant (*p* = 0.035). The mean duration of application for all patients was 32.13 ± 7.90 days, with an average duration of application of 30.27 days in the group with excellent grading, 33.64 days in the group with good grading, and 36.43 days in the group with moderate grading. The difference in the duration of application between the groups was statistically significant (*p* = 0.044), as shown in Table 5.

### 3.2. Systematic Review

#### 3.2.1. Success Rate

A forest plot for the success rate is shown in Figure 4. A total of nine studies were included in the classic method group, and six studies were included in the commercial product group for this comparison. The results were extracted using a random-effects model because of high heterogeneity. The classic method group showed a statistically smaller success rate, with a proportion of 0.79 (95% CI = 0.69 to 0.86), than the commercial product group, with a proportion of 0.83 (95% CI = 0.76 to 0.94).

#### 3.2.2. Complication Rate

A forest plot for the complication rate is shown in Figure 5. A total of six studies were included in the classic method group, and six studies were included in the commercial product group for this comparison. The results were extracted using a fixed-effects model in the classic method group because of low heterogeneity (I^2^ = 1%). A random-effects model was used because of high heterogeneity in the commercial product group (I^2^ = 85%). The classic method group showed a statistically smaller complication rate, with a proportion of 0.05 (95% CI = 0.03 to 0.10), than the commercial product group, with a proportion of 0.11 (95% CI = 0.06 to 0.19).

## 4. Discussion

Congenital ear deformities occur in 5% to 15% of infants [14,16]. Patients with congenital ear deformities tend to suffer psychological distress manifesting as anxiety, low self-esteem, and behavioral problems [12]. Traditionally, auricular deformities have been corrected surgically at 6 years of age [18]. Recently, nonsurgical management by ear molding has emerged as a favorable approach for newborn auricular deformities. Ear molding has shown much better outcomes in the correction of auricular deformities than otoplasty [13]. Ear molding not only spares operative morbidity but also allows for much earlier correction compared to surgical options, which usually address deformities only after the auricle has reached its adult size.

Commercial products are mostly used for ear molding of auricular deformities. However, costs for commercial products are expensive, placing them beyond the affordability of most parents from rural areas [6]. Therefore, the advantage of this nonsurgical intervention to correct auricular deformities is not possible for many infants. In contrast, the classic ear-molding method is much cheaper than commercial ear-molding products, and the materials needed for classic ear molding are easy to obtain. Because of these advantages, the classic ear-molding method can be another treatment option for patients whose families cannot afford commercial products.

It should be noted that commercial products are well-promoted, and their effectiveness has been intensively described in articles in the last 10 years. During this period, there was a lack of research on classic ear molding. Through this study, we confirmed the efficacy of the classic method of ear molding. In the case series, we achieved a high success rate in the correction of ear deformities using classic ear molding. Through a systematic review, we determined that the classic ear-molding method showed substantial efficacy in the treatment of newborn auricular deformities (79% and 87%) and a lower complication rate than commercial products (5% and 11%).

To achieve satisfactory outcomes, the timing of ear molding is most important [19]. The ideal intervention of auricular deformities is started within the first 2 weeks after birth. Infants older than 2 weeks show poor outcomes of correction and require a longer period of molding for successful correction [20,21]. Considering these points, it was presumed that early application of classic ear molding (5.81 ± 6.09 days) (Table 3) had an effect on the high success rate. Patient age at application and duration of molding were correlated with the outcome (*p* = 0.035 and *p* = 0.044, respectively) (Table 5). There was a difference in the average age at initiation in each group. This does not mean that earlier molding gives poorer results, but it may suggest that the appropriate timing of classic ear molding could be 4–5 days. However, this study was not designed to determine the appropriate timing of classic ear molding. There should be a lot of logical leaps to conclude that the appropriate timing of classic ear-molding could be 4–5 days. Further research is needed. Neither age at application nor duration of application had clinical significance because most patients started treatment within 2 weeks of birth. The type of auricle deformity is also a crucial factor affecting the efficacy [1]. However, there was no difference in the outcome of classic ear molding depending on the type of deformity in our study (*p* = 0.21) (Table 4).

There were several limitations in our study. First, this was a retrospective study. However, every attempt was made to minimize review bias by blinding reviewers to each other and clinical details. Second, there was a small number of patients with various types of deformities. This may have led to inaccurate statistical results, which need further analysis of large sample data to draw more scientific conclusions. Third, a short period of follow-up may be insufficient to conclude that the treatment outcomes can be maintained over time. Long-term follow-up demonstrated that obvious recurrent deformities did occur in some ears that initially were treated [22]. It is necessary to study whether classic ear molding could be effective for a long time. Fourth, the absence of objective tools for evaluating outcomes is another limitation of this study. Our data relied on independent plastic surgeon observations of the outcomes, which is not objective.

## 5. Conclusions

Ear molding is an effective nonsurgical treatment for ear anomalies and should be readily available and offered to parents [23]. Interspecialty collaboration is crucial to facilitate early identification and initiate prompt treatment [24]. Classic ear molding had remarkable and comparable therapeutic effects on neonatal auricular deformities. In addition, compared to other options, the classic ear-molding method is more suitable for infants with auricular deformities from socioeconomically vulnerable areas. Considering these advantages, the classic ear-molding method could be a better option for patients with congenital ear anomalies than commercial ear-molding products.

## Figures and Tables

**Figure 1 jcm-11-05751-f001:**
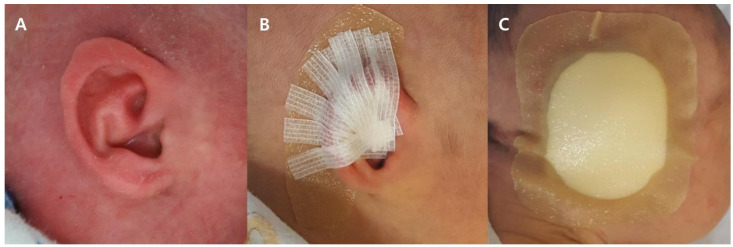
Treatment steps of classic ear molding. (**A**) Preparation before treatment. (**B**) Application of classic ear molding. (**C**) Protection of molding materials.

**Figure 2 jcm-11-05751-f002:**
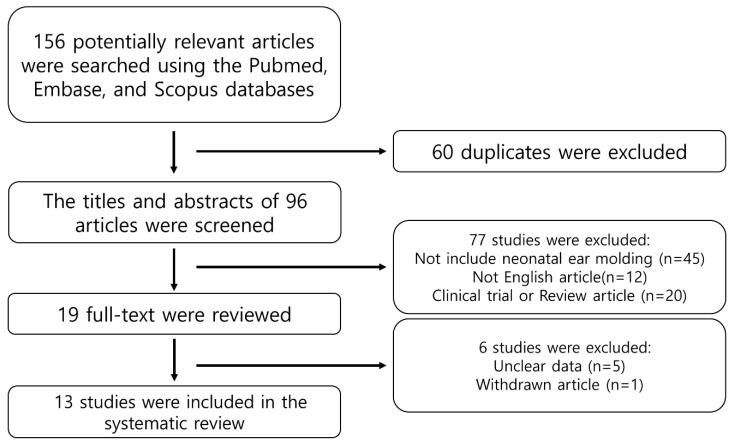
Flow diagram used to identify and select studies.

**Figure 3 jcm-11-05751-f003:**
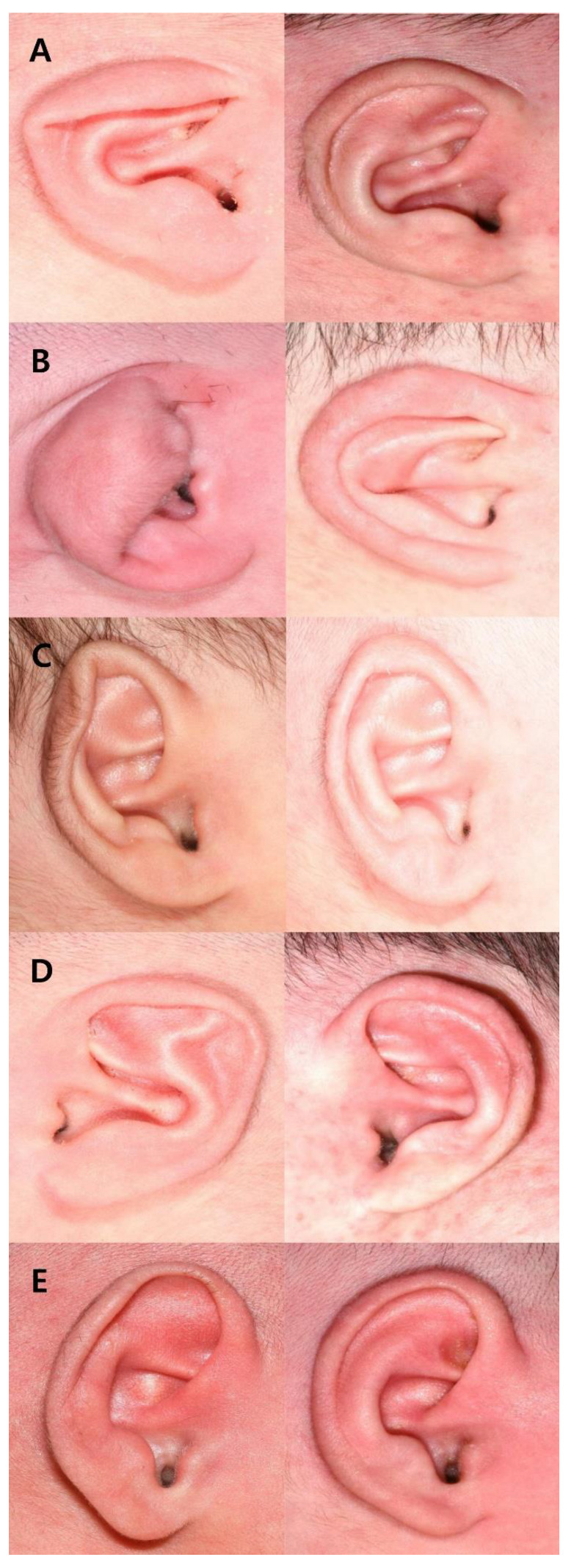
Typical results of classic ear molding for different types of ear deformities: (**A**) lop ear; (**B**) constricted ear; (**C**) helical rim deformity; (**D**) Stahl’s ear; (**E**) prominent ear.

**Figure 4 jcm-11-05751-f004:**
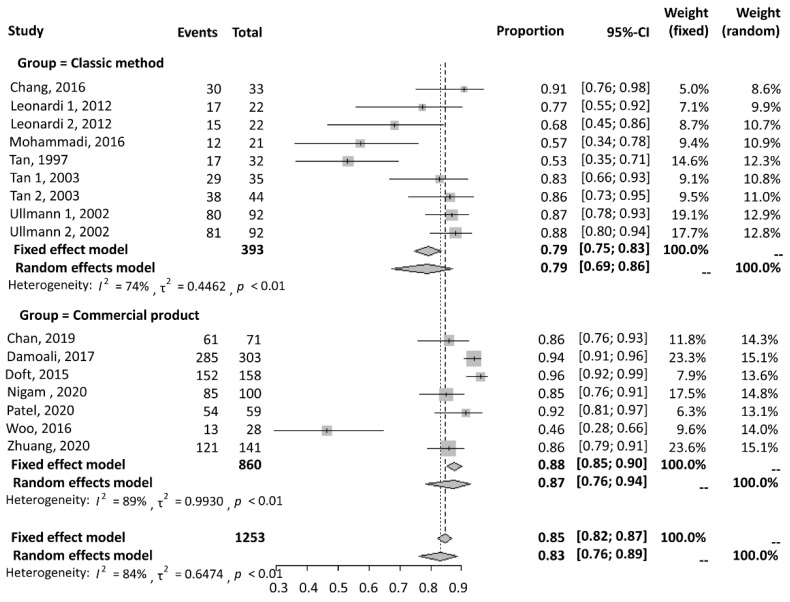
A forest plot of the success rate [4,7,8,9,10,11,12,13,14,15,16,17]. The classic ear-molding method showed substantial efficacy in the treatment of newborn auricular deformities.

**Figure 5 jcm-11-05751-f005:**
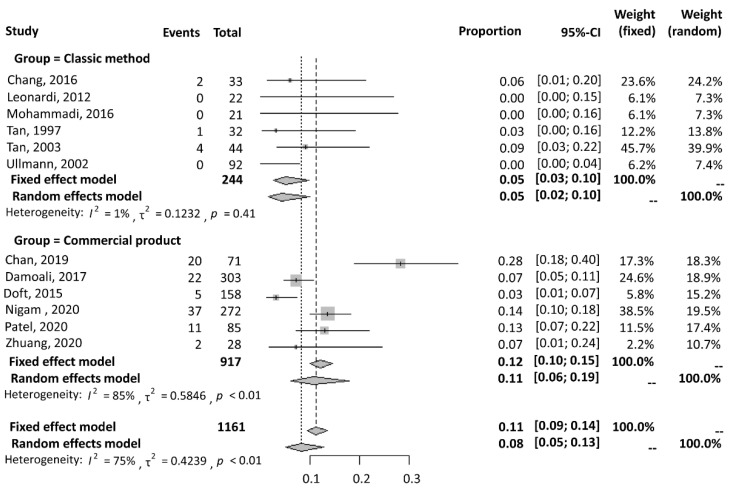
A forest plot of the complications [4,7,8,9,10,11,12,13,14,15,17]. The classic method showed a lower complication rate than commercial products.

**Table 1 jcm-11-05751-t001:** Photographic grading for ear anomalies.

Grade	Shape	Deformation/Malformation
Excellent	Normal ear shape	No appearance of original deformation/malformation
Good	Nearly normal ear shape	Mild yet nondistracting retention of original deformation/malformation
Moderate	Improved but not a normal ear shape	Noticeable, distracting retention of original deformation/malformation
Poor	No improvement	Abnormal ear shape with retention of original deformation/malformation

**Table 2 jcm-11-05751-t002:** Characteristics of the articles.

Classic Method									
Study No.	Study	Device	Cases (*n*)	Type of Anomaly (%)	Initiation (Mean Weeks ± SD; Range)	Duration (Mean Weeks ± SD; Range)	Responder	Success * (*n*)	Complication ^†^ (*n*)
1	Chang 2017 [7]	Conformer, Velcro tape, and gel sealing	33	Helical (30.3); Lop (18.2); Prom. (21.2); Stahl’s (24.2); mixed (6.1)	4.5; 0.86 to 15.1	3.9; 1.7 to 6.6	Physician	30	2
2	Leonardi ^1^ 2012 [8]	Splinting and taping	22	Const. (36.4); Lop (9.1); Prom. (18.2); Stahl’s (18.2)	N/A; 0.29 to 6	N/A; 5–8	Parents	17	0
	Leonardi ^2^ 2012 [8]						Physician	15	
3	Mohammadi 2016 [9]	Splinting and taping	21	Const. (38.1); Lop (23.8); Prom. (20.7)	7.52 ± 5.6; 2 to 24	13.33 ± 2; 11 to 18	Physician	12	0
4	Tan 1997 [4]	Splinting and taping	32	Concha (3.1); Lop (65.6); Prom. (18.8); Stahl’s (6.3); mixed (6.3)	2.4 ± 3.5; 0.14 to 10	10.9 ± 9; 5 to 38.5	Physician and parents	17	1
5	Tan ^1^ 2003 [10]	Splinting and taping	44	Cup (18.2); Lop (38.6); Helical (11.4) Prom. (31.8)	3.4; 0.14 to 15	7; 1 to 14	Parents	29 (total *n*: 35)	4
	Tan ^2^ 2003 [10]						Physician	38	
6	Ullmann ^1^ 2002 [11]	Splinting and taping	92	Costricted (21.7); Lop (30.4); Prom. (26.1); Stahl’s (21.7)	N/A; 0.14 to 1.43	6.8; 6 to 12	Parents	80	0
	Ullmann ^2^ 2002 [11]						Physician	81	
Commercial Product									
7	Chan 2019 [1]	Earwell^®^	71	Concha (0.95); Constricted (32.4); Cryptotia (0.95); Helical (9.5); Lidding (28.6); Lop (4.8); Prom.(4.8); Stahl’s (18.1)	2.24; 0 to 13.9	4.1; 1.5 to 6	Physician	61	20
8	Daniali 2017 [12]	Earwell^®^	303	Concha (36.4); Helical (24.8); Lidding (19.1); Prom. (8.9); Stahl’s (20.8)	1.79; N/A	5.29; 1.7 to 15.6	Physician	285	22
9	Doft 2015 [13]	Earwell^®^	158	Constricted (18); Cryptotia (0.5); Helical (38); Prom. (18.5); Stahl’s (25)	N/A; N/A	2; 1 to 6	Parents	152	5
10	Nigam 2020 [14]	Earwell^®^ /InfantEar^®^	272	Concha (2.2); Cup (7.7); Helical (14.3); Lidding/Lop (8.8); Prom. (23.2); Stahl’s (8.4)	2.9; N/A	N/A	Parents	85 (total *n*: 100)	37
11	Patel 2020 [15]	InfantEar^®^	85	Helical (100)	4.9; 8.6 to 10.9	4.4; 1 to 10.3	Survey	54 (total *n*: 59)	11
12	Woo 2016 [16]	Earwell^®^	28	Constricted (62.4); Cryptotia (7.1); Prom. (7.1); Stahl’s (21.4)	3.2; 0.7 to 7.4	4.7; 3.4 to 7.6	Parents	13	N/A
13	Zhuang 2020 [17]	Earlimn^®^	141	Concha (7.8); Constricted (17.7); Cryptotia (9.2); Cup (9.2); Helical (18.4); Lop (16.3); Prom. (10.6); Stahl’s (10.6)	5 ± 3.3; N/A	2.5 ± 1.86; N/A	Physician	121	2

Helical, helical deformity; Prom., prominent ear; Const., constricted ear; Concha, conchal deformity; * Normal to near normal ear shape; ^†^ skin irritation or superficial ulcer. The results of this divided to patients reported outcome and physician-reported outcome. ^1^ is for patients outcome and ^2^ is for physician outcome.

**Table 3 jcm-11-05751-t003:** Summary of ear anomalies.

		*n*	%
Sex	M	41	43.6
	F	53	56.4
type	Lop ear	65	69.1
	Constricted ear	12	12.8
	Helical rim deformity	7	7.4
	Stahl’s ear	7	7.4
	Prominent ear	3	3.2
Grade	Excellent	48	51.1
	Good	39	41.5
	Moderate	7	7.4
	Poor	0	0
Age at application(days)		5.81 ± 6.09	6.09
Duration of application(days)		32.13 ± 7.90	7.9

**Table 4 jcm-11-05751-t004:** Types of anomalies and outcomes.

	Grade					*p*
	Excellent	Good	Moderate	Poor	Total	
Lop ear	35(54)	26(40)	4(6)	0(0)	65(100)	0.210
Constricted ear	3(25)	7(58)	2(17)	0(0)	12(100)	
Helical rim deformity	2(29)	4(57)	1(14)	0(0)	7(100)	
Stahl’s ear	6(86)	1(14)	0(0.0)	0(0)	7(100)	
Prominent ear	2(67)	1(33)	0(0.0)	0(0)	3(100)	
Total	48	39	7	0(0)	94	

**Table 5 jcm-11-05751-t005:** Comparison of outcome grading with duration and age at application.

	Excellent	Good (*n* = 39)	Moderate (*n* = 7)	Total (*n* = 94)	*p*
	(*n* = 48)				
Age at application (days)					
Mean ± SD	4.58 ± 4.74	7.72 ± 7.28	3.71 ± 4.64	5.82 ± 6.09	0.035 ^a^
Median(range)	4 (1–22)	4 (1–22)	2 (1–22)	4 (1–22)	0.007 ^b^
Duration of application (days)					
Mean ± SD	30.27 ± 8.58	33.64 ± 7.07	36.43 ± 2.94	32.13 ± 7.90	0.044 ^a^
Median(range)	35 (15–47)	35 (15–47)	35 (15–47)	35 (15–47)	0.159 ^b^

^a^: one way ANOVA. ^b^: Kruskal Wallis test.

## Data Availability

The data presented in this study are available on request from the corresponding author.

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
