# Peer review of "Efficacy of Classic Ear Molding for Neonatal Ear Deformity: Case Series and Literature Review"

_jcm, 2022, doi:10.3390/jcm11195751_

Round 1
Reviewer 1 Report
It is of interest to compare the classic ear-molding with current commercial products. However, the author's description of the classic ear-molding method is too simple, and the readers can not get accurate information about the classic ear-molding. It is suggested that the author provide more detailed description of the method or picture explanation. Not like “The classic ear-molding materials are custom-made splints of metal wire, surgical 62 tape, foam, and silicone tape.”
Author Response
It is of interest to compare the classic ear-molding with current commercial products. However, the author's description of the classic ear-molding method is too simple, and the readers can not get accurate information about the classic ear-molding. It is suggested that the author provide more detailed description of the method or picture explanation. Not like “The classic ear-molding materials are custom-made splints of metal wire, surgical 62 tape, foam, and silicone tape.”
Response: We completely agree with your comment. We newly added an image of the treatment steps of classic ear molding. It could be helpful to understand our technique.
Figure 1. Treatment steps of classic ear molding
Preparation before treatment (B) Application of classic ear molding (C) Protection of molding materials

Reviewer 2 Report
I read manuscript by Kim et al. with pleasure and great interest. This manuscript compares treatment methods of ear deformities in newborns using the classic ear molding with molding with use of commercial products and additionally presents their own results. Below the authors may find some comments/suggestions that may improve the quality of the article:
Material and methods:
Line 60 (page2) It might be interesting to image the treatment steps. There are a lot of tutorials of use the commercial ear molding product and lack of classic ear molding visualization - it could be interesting to see authors technique.
Line 25 Figure 2 is missing photo with helical rim deformity and treatment result.
Line 37 and 96. Patient and duration of application: “the 37 average age at initiation was 4.58 days in the excellent group, 7.72 days in the good group, 38 and 3.71 days in the moderate group. This was statistically significant (p = 0.035)”. Does it mean that the earlier molding gives poorer results? Do the authors suggest the right time (4-5 days) for molding? Or there are other reasons of this results?
Line 60. Complication rate: Correct the sentence “A forest plot for the success rate is shown in Figure 4”
Discussion:
It is especially important to discuss the types of complications. For example, the superficial excoriation is the most common and it is easy to treat with good healing after 5 days of treatment. The other, for instance allergic reactions or infection will terminate the treatment and will implicate the result. It could be good to compare severe complications and those excluding further molding treatment.
Line 109. When authors dispute about limitations of the study and the long-time results, it might be interesting to cite:
Zhong Z, Zhang J, Xiao S, Liu Y, Zhang Y. Long-Term Effectiveness of Ear Molding in Infants Using the EarWell Infant Correction System in China. Plast Reconstr Surg. 2021 Sep 1;148(3):616-623. doi: 10.1097/PRS.0000000000008293. PMID: 34432691.
There are some words errors:
Line 38 “birth and decrease to nadir six weeks after birth”
Figure 2. Typical results of classic ear molding for different types of ear deformities (A) Lop ear (B) Contricted ear (C) Stahl’s ear (D) Prominent ear
To sum it up it is really interesting manuscript showing similar effectiveness of classic ear molding to molding using ready-made sets. Based on work of Zhong, it should be assumed that the effectiveness decreases in time, however, this should be still considered as high.
It should be noted that commercial products are very promoted and their effectiveness has been intensively described in articles in the last 7 years. During this period, there is a lack of research on classic ear molding and the current article is a response to this need. Authors draws attention to the beneficial economic aspect of classic ear molding and, in my opinion, demonstrating a similar number of important complications would promote the use of the classical method.
Author Response
Thank you for the constructive review of our manuscript and for the detailed reviewer comments. Accordingly, we have revised the manuscript based on the comments provided. Detailed responses to each comment are outlined below.
Reviewer 2
I read manuscript by Kim et al. with pleasure and great interest. This manuscript compares treatment methods of ear deformities in newborns using the classic ear molding with molding with use of commercial products and additionally presents their own results. Below the authors may find some comments/suggestions that may improve the quality of the article:
Thank you for your favorable comment on our study.
Material and methods:
Line 60 (page2) It might be interesting to image the treatment steps. There are a lot of tutorials of use the commercial ear molding product and lack of classic ear molding visualization - it could be interesting to see authors technique.
Response: I apologize that we did not provide a detailed method of classic ear molding. We added an image of the treatment steps of classic ear molding.
Line 25 Figure 2 is missing photo with helical rim deformity and treatment result.
Response: I apologize for the lack of specific figures. We added a photo of the helical rim deformity (before correction and after molding) in Figure 3. The treatment outcome of helical rim deformities was also excellent.
Line 37 and 96. Patient and duration of application: “the 37 average age at initiation was 4.58 days in the excellent group, 7.72 days in the good group, 38 and 3.71 days in the moderate group. This was statistically significant (p = 0.035)”. Does it mean that the earlier molding gives poorer results? Do the authors suggest the right time (4-5 days) for molding? Or there are other reasons of this results?
Response: We are very grateful for your comment. To achieve satisfactory outcomes, the timing of ear molding is most important. The ideal intervention of auricular deformities is started within the first 2 weeks after birth. In our case study, most patients were treated within 2 weeks, and I think this may have affected our good treatment outcomes. There was a difference in the average age at initiation in each group. This does not mean that earlier molding gives poorer results but it may suggest that the appropriate timing of classic ear molding could be 4-5 days. However, this study was not designed to determine the appropriate timing of classic ear molding. There should be a lot of logical leaps to conclude that the appropriate timing of classic ear-molding could be 4-5 days. Further research is needed. We revised the phrase you mentioned to be more understandable.
Line 60. Complication rate: Correct the sentence “A forest plot for the success rate is shown in Figure 4”
Response: I apologize that we did not write this correctly. I corrected the error.
Discussion:
It is especially important to discuss the types of complications. For example, the superficial excoriation is the most common and it is easy to treat with good healing after 5 days of treatment. The other, for instance allergic reactions or infection will terminate the treatment and will implicate the result. It could be good to compare severe complications and those excluding further molding treatment.
Response: Thank you for your comment. We completely agree with your comment. We did not provide detailed information on complications to support our argument. In the case series, five of 94 ears (5.3 percent) experienced complications during the treatment period, which included dermatitis and skin excoriations. However, we did not compare the complication rate by type and severity in the systemic review. There was no definite evidence that directly proved that classic ear molding could have a lower severity of complications. Further research is needed. We revised the phrase about complications in the case series results & discussion.
Line 109. When authors dispute about limitations of the study and the long-time results, it might be interesting to cite:
Zhong Z, Zhang J, Xiao S, Liu Y, Zhang Y. Long-Term Effectiveness of Ear Molding in Infants Using the EarWell Infant Correction System in China. Plast Reconstr Surg. 2021 Sep 1;148(3):616-623. doi: 10.1097/PRS.0000000000008293. PMID: 34432691.
Response: Thank you for your comment. We read the paper you mentioned. Long-term follow-up demonstrated that obvious recurrent deformities did occur in some ears that initially were treated. I checked the limitation through this paper and this point clearly needs to be reevaluated. We quoted this paper in our study. Thank you again for your comment.
There are some words errors:
Line 38 “birth and decrease to nadir six weeks after birth”
Figure 2. Typical results of classic ear molding for different types of ear deformities (A) Lop ear (B) Contricted ear (C) Stahl’s ear (D) Prominent ear
Response: I apologize that we did not write these correctly. I corrected the errors.
To sum it up it is really interesting manuscript showing similar effectiveness of classic ear molding to molding using ready-made sets. Based on work of Zhong, it should be assumed that the effectiveness decreases in time, however, this should be still considered as high.
It should be noted that commercial products are very promoted and their effectiveness has been intensively described in articles in the last 7 years. During this period, there is a lack of research on classic ear molding and the current article is a response to this need. Authors draws attention to the beneficial economic aspect of classic ear molding and, in my opinion, demonstrating a similar number of important complications would promote the use of the classical method.
Thank you for your favorable comment on our study.

Reviewer 3 Report
Congratulations to the authors on their work. The submission consists of a case series presentation and a literature review on classic molding in neonates with ear malformations.
A few suggestions:
- The clinical study has to be defined better. What type of study was it - prospective or not? What were the exclusion criteria, if any? Who decided the splinting total duration for each patient and how? How often were the patients monitored during splinting? Were the authors able to isolate any specific factors that might predict or influence the failure of the conservative treatment? What complications, if any, did the patients have?
- The authors should add some photos of the splints they manufactured and used.
- In the Discussions section, the authors state, "To the best of our knowledge, this is the first study to compare the effects of the classic ear-molding method and commercial ear-molding products. Through this study, we confirmed the efficacy of the classic method of ear molding. In the case series, we achieved a high success rate in the correction of ear deformities using classic ear molding". This paragraph is confusing because there is no previous mention in the study's description of any comparison with commercial ear-molding products.
- The limitations of the study are hefty. The authors acknowledge them honestly. Still, the article's conclusions disregard entirely said limitations. The authors should scale their enthusiasm and reformulate this section accordingly.
Author Response
Thank you for the constructive review of our manuscript and for the detailed reviewer comments. Accordingly, we have revised the manuscript based on the comments provided. Detailed responses to each comment are outlined below.
Reviewer 3
Congratulations to the authors on their work. The submission consists of a case series presentation and a literature review on classic molding in neonates with ear malformations.
Thank you for the review of our study.
A few suggestions:
The clinical study has to be defined better. What type of study was it - prospective or not?
Response: I apologize that we did not make this clearer. Our study is a retrospective study, which is one of the limitations of our paper. However, every attempt was made to minimize review bias by blinding reviewers to each other and clinical details. We revised the phrase about our study’s limitation in the discussion.
What were the exclusion criteria, if any? Who decided the splinting total duration for each patient and how? How often were the patients monitored during splinting?
Response: I apologize that we did not make this clearer. A retrospective review of 54 consecutive infants treated with classic ear molding in our hospital from January 2019 to March 2022 was performed. When we first started treatment, patients visited the clinic to check for complications at three days after ear-molding. After we confirmed that five patients did not have any complications at three days after treatment, all infants were scheduled for weekly follow-up routinely to monitor complication and auricle changes and to modulate the devices to obtain better correcting conditions. The infants wore the splint for 24 hours per day.
For the first week of treatment, taping was used to counteract the significant resistive forces to expansion resulting from tissue deficiency. After that, the splint was curved according to the desired shape of the helix and was then fixed on the anterolateral surface of the auricle. Splinting was performed continuously until 1 week after normalization of the auricular anatomy was achieved, and it was stopped after a maximum of 6 weeks of treatment even when no correction occurred. We revised the phrase about treatment in the Methods section.
Were the authors able to isolate any specific factors that might predict or influence the failure of the conservative treatment? What complications, if any, did the patients have?
Response: I apologize that we did not make this clearer. To achieve satisfactory outcomes, the timing of ear molding is most important. The ideal intervention of auricular deformities is started within the first 2 weeks after birth. In our case study, most patients were treated within 2 weeks, and I think this may have affected our good treatment outcomes.
In the case series, five of 94 ears (5.3 percent) experienced complications during the treatment period, which included dermatitis and skin excoriations. When skin lesions occurred, we continued the treatment because foam-covered molding materials absorb exudate and improve skin lesions. The splinting would only have stopped when pressure ulcers occurred, but they did not occur. We revised the phrase about complications in the case series results & discussion.
The authors should add some photos of the splints they manufactured and used.
Response: I apologize that we did not provide a detailed method of classic ear molding. We added an image of the treatment steps of classic ear molding.
In the Discussions section, the authors state, "To the best of our knowledge, this is the first study to compare the effects of the classic ear-molding method and commercial ear-molding products. Through this study, we confirmed the efficacy of the classic method of ear molding. In the case series, we achieved a high success rate in the correction of ear deformities using classic ear molding". This paragraph is confusing because there is no previous mention in the study's description of any comparison with commercial ear-molding products.
Response: I apologize that we did not make this clearer. We revised the phrase you mentioned to a more understandable one as follows.
“It should be noted that commercial products are very promoted and their effectiveness has been intensively described in articles in the last 10 years. During this period, there was a lack of research on classic ear molding. Through this study, we confirmed the efficacy of the classic method of ear molding. In the case series, we achieved a high success rate in the correction of ear deformities using classic ear molding.”
The limitations of the study are hefty. The authors acknowledge them honestly. Still, the article's conclusions disregard entirely said limitations. The authors should scale their enthusiasm and reformulate this section accordingly.
Response: We apologize for the missing limitations of our study in the Discussion section, which is a basic and essential part of the article. We added the limitations of the study to the Discussion section.

Round 2
Reviewer 1 Report
According to the suggestion, the author added photos to explain the classical method, which makes the article easier to understand. I think the comparative study of different method is still very necessary, and I suggest publication
Reviewer 3 Report
I would like to thank the authors for the provided explanations and congratulate them again on their work.